# Parasites and RNA viruses in wild and laboratory reared bumble bees *Bombus pauloensis* (Hymenoptera: Apidae) from Uruguay

**Sheena Salvarrey**[1]*, **Karina Antúnez**[2], **Daniela Arredondo**[2], **Santiago Plischuk**[3], **Pablo Revainera**[4], **Matías Maggi**[4], **Ciro Invernizzi**[1]

**1** Facultad de Ciencias, Sección Etología, Montevideo, Uruguay, **2** Departamento de Microbiología, Instituto de Investigaciones Biológicas Clemente Estable (IIBCE), Montevideo, Uruguay, **3** Centro de Estudios Parasitológicos y de Vectores (CEPAVE) (CONICET- UNLP), La Plata, Argentina, **4** Centro de Investigación en Abejas Sociales (CIAS), Facultad de Ciencias Exactas y Naturales, Mar del Plata, Argentina

* ssalvarrey@fcien.edu.uy

## Abstract

Bumble bees (*Bombus* spp.) are important pollinators insects involved in the maintenance of natural ecosystems and food production. *Bombus pauloensis* is a widely distributed species in South America, that recently began to be managed and commercialized in this region. The movement of colonies within or between countries may favor the dissemination of parasites and pathogens, putting into risk while populations of *B. pauloensis* and other native species. In this study, wild *B. pauloensis* queens and workers, and laboratory reared workers were screened for the presence of phoretic mites, internal parasites (microsporidia, protists, nematodes and parasitoids) and RNA viruses (Black queen cell virus (BQCV), Deformed wing virus (DWV), Acute paralysis virus (ABCV) and Sacbrood virus (SBV)). Bumble bee queens showed the highest number of mite species, and it was the only group where Conopidae and *S. bombi* were detected. In the case of microsporidia, a higher prevalence of *N. ceranae* was detected in field workers. Finally, the bumble bees presented the four RNA viruses studied for *A. mellifera*, in proportions similar to those previously reported in this species. Those results highlight the risks of spillover among the different species of pollinators.

## Introduction

Wild and managed pollinators are essential for agricultural production, maintenance of biodiversity and the sustainability of natural ecosystems [1–3]. However, they are threatened by different factors including intensification of land use, intoxication with pesticides or infection by multiple pest and pathogens, among others [2]. In particular, wild bumble bees populations of the genus *Bombus* (Hymenoptera: Apidae), are in global decline [4,5]. Among the main threats for bumble bee health, different parasitic enemies stand out, some of them are specific to the

**Funding:** This study was funded by the Agencia Nacional de Investigación e Innovación (ANII) (register number: POSNAC_2014_1_102699) and Comisión Académica de Posgrado (CAP) of the Universidad de la República (Udelar) through PhD scholarship to S.S. Comisión Sectorial de investigación Científica (CSIC, Udelar) also provided support through a movility program.

**Competing interests:** The authors have declared that no competing interests exist.

genus *Bombus*, while others have a broad host spectrum [6,7]. The extended commerce and movement of managed bees, such as honey bees *Apis mellifera* L. and some bumble bees, has led to the spread of pathogens to new hosts, a phenomenon known as spillover [6,8–11].

The microsporidium *Nosema ceranae* Fries [recently Tokarev *et al.* [12] suggest to be reclassified as *Vairimorpha ceranae*)] is one of the most documented spillover examples. This parasite is found in honey bees [13], bumble bees [14–17], stingless bees [18,19], solitary bees (Euglossini) [20] and social wasps [18]. Another example of pathogen spillover occurs with RNA viruses of *A. mellifera*. Acute bee paralysis virus (ABPV), Black queen cell virus (BQCV), Deformed wing virus (DWV) and Sacbrood virus (SBV) [21,22] were described in honey bees but have also been found in bumble bees [10,23,24], stingless bees [25,26], carpenter bees [27] and other insects such as syrphids (Diptera) [28] and butterflies (Lepidoptera) [29]. Honey bees colonies acting as reservoirs, facilities the spread of pathogens and viruses to other pollinator species through the flowers they share [9,24,30].

*Bombus pauloensis* Friese (= *Bombus atratus*] is widely distributed throughout South America [31,32], and is utilized successfully in production of tomato (*Solanum lycopersicum* L.) and pepper (*Capsicum annum* L.) in greenhouses [33–35], as well as that of red clover (*Trifolium pratense* L.) seeds [36]. The colonies of *B. pauloensis* have been raised in captivity in small scale both in Colombia and Uruguay [36,37] and at commercial scale in Argentina, following a very extended breeding practice of some European and North American species [38,39].

*Bombus pauloensis* is distributed throughout the Uruguayan territory, and alongside *Bombus bellicosus* Smith, whose distribution is more reduced, are the only two *Bombus* species found in the country [40]. Previous studies reported the presence of internal and external parasites in queens, workers and males of both species, including the microsporidia *N. ceranae* [16,41] and *Tubulinosema pampeana* Plischuk et al., the nematode *Sphaerularia bombi* Dufour, one species of parasitoid diptera [41], and the external mites, *Kuzinia* spp. Zachvatkin, *Pneumolaelaps longanalis* Hunter and Husband, *Pneumolaelaps longipilus* Hunter, *Scutacarus acarorum* Goeze, and *Tyrophagus putrescentiae* Schrank [42].

In the southern region of South America (Argentina and Chile) the dispersion of the exotic species *Bombus terrestris* L. and *Bombus ruderatus* F., has put under threat native bumble bees species, as *Bombus dahlbomii* Guérin-Méneville, which is currently endangered [43–46]. These invasive species, introduced in Chile over the last few years, may have been acting as reservoirs of pathogens that jumped to native species causing significant damage [43,46,47]. Uruguay, as a neighbor country of Argentina, is also under risk of invasion by *B. terrestris* or *B. ruderatus*, or even by pathogens originally present on these species than now had spread to some South American native species [43–47].

From a sanitary point of view, the artificial breeding conditions (high density of individuals, impossibility of going out to defecate and forage, and limited food availability), can increase the survival and multiplication of different pathogens, facilitating the proliferation and transmission of diseases [8,48]. Thus, the aim of this study was to evaluate the presence of different parasites and pathogens on wild *B. pauloensis* queens and workers; and to evaluate if artificial breeding condition can increase infection by pathogens.

## Materials and methods

### Bumble bee collection

During the spring (September 2014), 73 queens of *B. pauloensis* were collected while foraging in the Faculty of Agronomy, University of the Republic, Montevideo (34˚ 50' S, 56˚ 13' W) after finishing their hibernation period. Among these, 19 queens were used for parasite analysis, 14 for viral analysis and 40 to start laboratory rearing according to Salvarrey *et al.* [36].

When the laboratory colonies reached 25 workers, a total of 92 were collected from 10 different colonies. Among these, 46 were used for parasite analysis and 46 for viral analysis.

When wild workers started to emerge in nature, in autumn (March 2015), 54 wild workers were collected in the same area as the queens, from which 37 were used for parasite analysis and 17 for virus analysis. The individuals used for parasite analysis were kept at -20˚C, and those assigned to virus analysis were kept at -80˚C.

## Identification of mites and internal parasites

In order to detect phoretic mites, individual bumble bees were observed with a magnifying glass (40x). The mites were extracted, separated and observed with a light microscope (400X) for identification using taxonomic keys [49–52].

Prevalence (percentage of bumble bees harboring mites), abundance (number of mites per examined bumble bee), and intensity (number of mites per parasitized bumble bee) was determined for each mite species and in the three bumble bee groups (wild queens and workers, and laboratory reared workers).

Besides that, mite diversity per group was calculated using Simpson's index. Results were expressed as low (0–0.3), moderate (0.3–0.6) and high (0.6–1) diversity, according to Revainera *et al.* [42].

For internal parasite identification, bumble bees were dissected under a stereoscopic microscope (10x – 40x). Firstly, the metasomal cavity was thoroughly observed looking for nematodes and diptera larvae, and trachea was scrutinized in search of mites [41]. Then, small samples of fat tissue, Malphigian tubules, midgut and posterior intestine were extracted and observed under compound bright field microscope (400x - 1000x) in order to detect microsporidia and protists (*e.g.* Kinetoplastidea, Neogregarinorida) [53]. Special attention was given to the fat tissue since the abnormal presence of granules in this tissue could be provoked by the presence of *T. pampeana* [54]. In the cases in which microsporidia was observed, the body of the infected insects was completely homogenized using 2 ml of distilled water and the number of microsporidia spores was quantified using a Neubauer chamber [55].

## Detection of RNA viruses

Workers and queens samples were individually placed in 1.5 ml tubes and 500 µl or 1200 µl of PBS, respectively, were added. Individuals were disrupted and homogenized using a sterile glass rod. Total RNA was isolated from each individual bee using the PureLink® Viral RNA/ DNA Mini Kit (Invitrogen™). Co-purified DNA was degraded usingDNase I, Amplification Grade (Invitrogen™), according to the manufacturer´s recommendations. Then the reverse transcription to cDNA was performed using the High Capacity cDNA Reverse Transcription Kit (Applied Biosystems™, EEUU), according to the manufacturer´s instructions. Viral detection was carried out by real time PCR using Power SYBR® Green PCR Master Mix (Applied Biosystems, EEUU) and specific *primers* for reference and viral genes (Table 1). Reaction mixture consisted of 1X Master Mix, 0.5 µM of each *primer*, RNAse free water and 5 µl of 1:10 diluted cDNA in a final volume of 25 µl. Negative controls were included on each run. Serial dilutions of a mix of all the samples were used as a standard curve.

Real time PCR reactions were carried out in a thermal Bio-Rad CFX96 Touch ™ Real-Time System (Bio-Rad, USA). The cycling program consisted of an initial activation at 95˚C for 10 minutes, and 40 cycles of 95˚C for 15 seconds, 50˚C for 30 seconds and 60˚C for 30 seconds.

**Table 1. Primers utilized for the quantification of viruses in the samples through qPCR.**

| Primer | Sequence 5'– 3' | Virus/Gen | Reference |
|--------|-----------------|-----------|-----------|
| ABPV1 | ACCGACAAAGGGTATGATGC | ABPV | Johnson *et al.*, 2009 |
| ABPV2 | CTTGAGTTTGCGGTGTTCCT | | |
| DWV_F | CTGTATGTGGTGTGCCTGGT | DWV | Kukielka *et al.*, 2008 |
| DWV_R | TTCAAACAATCCGTGAATATAGTGT | | |
| BQCV_F | AAGGGTGTGGATTTCGTCAG | BQCV | Kukielka *et al.*, 2008 |
| BQCV_R | GGCGTACCGATAAAGATGGA | | |
| SBV_F | GGGTCGAGTGGTACTGGAAA | SBV | Johnson *et al.*, 2009 |
| SBV_R | ACACAACACTCGTGGGTGAC | | |
| BACTIN1 | ATGCCAACACTGTCCTTTCTGG | β-actina | Yang & Cox-Foster, 2005 |
| BACTIN2 | GACCCACCAATCCATACGGA | | |

The specificity of the reaction was verified through the inclusion of a melting curve of the amplified products (from 65 to 95˚C). The β-actin mRNA was amplified in each sample as a control of correct RNA manipulation and extraction.

## Statistical analyses

The prevalence of the different pathogens and mites in the wild queens and workers and laboratory reared workers was compared using the Chi-square test. The intensity of the infection by microsporidia as well as the number of mites in the three groups of bumble bees were compared using the Kruskal Wallis and Mann-Whitney tests. P values under 0.05 were considered significant. Statistical analyses were performed using INFOSTAT (available at http://www. infostat. com. ar).

## Results

Multiple parasites and pathogens were identified on bumble bees, including the mites *T. putrescentiae*, *P. longanalis*, *P. longipilus*, *Kuzinia* sp. and *Parasitellus fucorum*, the microsporidia *N. ceranae* and *T. pampeana*, a diptera of Conopidae family, the nematode *S. bombi*, and the RNA viruses BQCV, ABPV, SBV and DWV. Other common bumble bee parasites such as *Apicytis* sp. and *Chritidia bombi* were not found.

## Phoretic mites

Fifty eight percent of the screened bumble bees were infected by at least one species of mite. The prevalence was higher in the queens (73.6%), followed by the laboratory workers (65.2%) and the wild workers (40.5%) ($\chi^2 = 7.52$; $p = 0.02$; df = 2). The most frequently found mites were *T. putrescentiae*, which was detected mainly in queens and laboratory workers (H = 21.56; P< 0.0001) and *Kuzinia* sp. in the wild workers (H = 15.36; P< 0.0001) (Table 2).

Queens were infested by the highest number of mite species ($\chi^2 = 12.89$; p = 0.0016; df = 2) and 64.3% of them showed between two and four species of mites per individual. Co-infestation was less observed in wild workers or in laboratory workers (13.3% and 16.6%, respectively).

According to Simpsons' diversity index over 90% of bumble bees of all groups had low diversity of mites (Fig 1). On the other hand, mean values of the index were 0.18 for the queens, 0.12 for the wild workers and 0.08 for the lab worker bees.

**Table 2. Prevalence (P), abundance (A) and intensity (I) of the observed mites on laboratory workers, wild workers and queens of *B. pauloensis*.**

| Bumble bee group | | *T. putrescentiae* | *P. longanalis* | *P. longipilus* | *Kuzinia* spp. | *P. fucorum* |
|---|---|---|---|---|---|---|
| Laboratory workers (N = 46) | P | **58.7** | - | 8.7 | 13.0 | - |
| | A | 4.0 | - | 0.0 | 0.2 | - |
| | I | 6.7 | - | 1.0 | 1.3 | - |
| Wild workers (N = 37) | P | 10.8 | 2.7 | 2.7 | **29.7** | 2.7 |
| | A | 0.2 | 0.1 | 0.0 | 3.6 | 0.0 |
| | I | 1.8 | 4.0 | 1.0 | 1.3 | 1.0 |
| Queens (N = 19) | P | **63.2** | **31.6** | 26.3 | 21.1 | - |
| | A | 26.5 | 2.7 | 0.3 | 0.3 | - |
| | I | 42.0 | 8.7 | 1.2 | 1.3 | - |
| Total (N = 102) | P | 42.2 | 6.9 | 9.8 | 20.6 | 1.0 |
| | A | 6.8 | 0.5 | 0.1 | 1.4 | 0.0 |
| | I | 16.1 | 8.0 | 1.1 | 7.0 | 1.0 |

The highest prevalence values are shown in black.

## Internal pathogens

The microsporidia *N. ceranae* and *T. pampeana* were found in the three analyzed groups of bumble bees. Twenty six percent of the screened bumble bees were infected by *N. ceranae*. Its prevalence was higher in wild workers (45.9%) than in laboratory workers (13%) ($\chi^2$ = 12.6; p = 0.0004; df = 1) and in queens (16.6%) ($\chi^2$ = 5.78; p = 0.01; df = 1). The prevalence values of the last two groups were similar ($\chi^2$ = 0.08; p = 0.77; df = 1). Regarding the intensity of the infections, similar values were found in the three groups ($U$ = 1.58; p = 0.45). Queens showed 2.1 ± 2.9 x 10$^6$ spores/bee, wild workers 2.4 ± 0.96 x 10$^5$ spores/bee and the laboratory workers 3.4 ± 4.8 x 10$^5$ spores/bee.

Almost fourteen percent of the bumble bees were infected with *T. pampeana* (21% of the queens, 8.1% of the wild workers, 15.2% of the laboratory workers). No significant differences were found in the prevalence per groups ($\chi^2$ = 1.93; p = 0.38; df = 2). Regarding the intensity of the infections with this microsporidium, queens showed 6.4 ± 2.5 x 10$^5$ spores/bee, wild

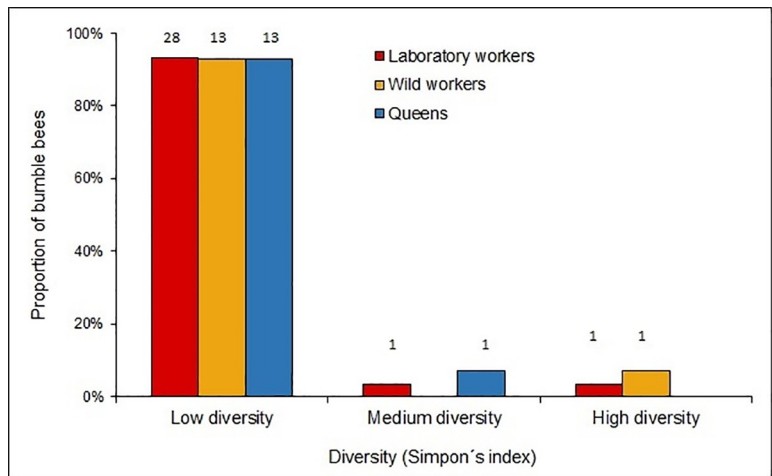

**Fig 1. Diversity of phoretic mites in bumble bees *B. pauloensis*.** Proportion of bumble bees with low (0–0.3), moderate (0.3–0.6) and high (0.6–1) diversity of phoretic mites based on Simpson's index values. Numbers above columns indicate sample size.

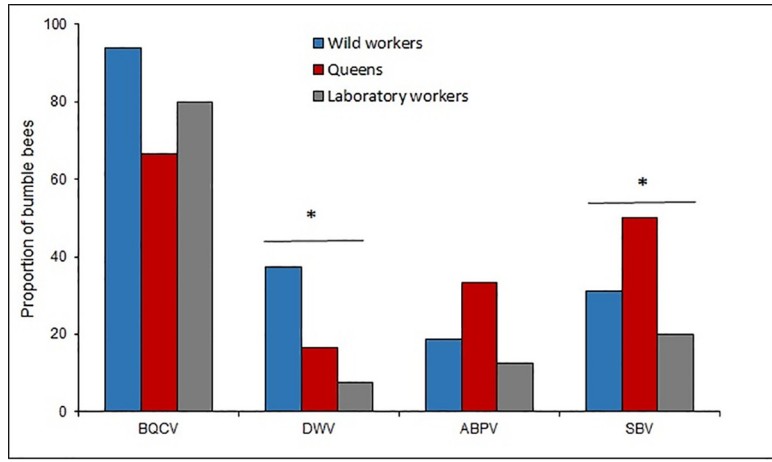

**Fig 2. Prevalence of BQCV, DWV, ABPV and SBV in laboratory workers, wild workers and queens.** The asterisk* indicates significante differences (P < 0.05) between the bumble bee groups for the Chi-square test.

workers 5.2 ± 8.0 x $10^5$ spores/bee and laboratory workers 2.4 ± 3.5 x $10^5$ spores/bee, with no differences between groups ($U$ = 2.86; p = 0.23).

Three cases of coinfection (2.9%) with both types of microsporidia were found, two in wild workers and one in a queen.

## Parasites

The nematode *S. bombi* was found in two of the 19 analyzed queens (10.5%), counting a total of seven gravid females (hypertrophied uteri) in one of them, and two in the other.

## Parasitoids

Diptera larvae belonging to Conopidae family were found in six wild workers (16.2%, n = 37) and in two queens (10.5%, n = 19) ($\chi^2$ = 7.69; p = 0.02; df = 2). No parasitoids were found in laboratory workers.

## RNA viruses

In 83.8% of the analyzed bumble bees at least one RNA virus was detected. BQCV was the most prevalent virus (80.9% of the samples), while SBV, DWV and ABPV showed lower values (Fig 2). Regarding the detection of virus among the analyzed groups, differences were found for the DWV ($\chi^2$ = 7.59; p = 0.02, df = 2) and for the SBV ($\chi^2$ = 4.65; p = 0.09; df = 2), since those were more prevalent in wild workers and queens, respectively (Fig 2).

Of the analyzed bumble bees 55.8% were only infected by one virus, mainly BQCV; while 27.9% showed co-infection with different viruses, including BQCV-ABPV (n = 9), BQCV-DWV (n = 4), BQCV-SBV (n = 3) and ABPV-SBV (n = 1). Triple infection was found in two samples, with only one case found in laboratory workers and in queens.

## Discussion

Bumble bee colonies have an annual life cycle and only the queens survive the winter. This factor has shaped the behavior of parasites and pathogens to reproduce and spread beyond the period in which colonies disappear [56–58].

The effect of phoretic mites in bumble bee populations is unclear. Many groups feed on wax and pollen, while others consume small nematodes and fungi, which might be beneficial for bumble bees [59–61]. However, mites can act as vectors facilitating the introduction of fungi and pathogens. In this sense, Revainera *et al.* [62] found in individuals of both *P. longanalis* and *P. fucorum* obtained from bumble bees collected since 1940's, the presence of *Ascosphaera* spp., *N. ceranae*, *Nosema apis*, and *Nosema bombi*, *Crithidia bombi*, *Lotmaria passim* (Euglonozoa; Trypanosomatidae), *Apicystis bombi* (Apicomplexa: Neogregarinorida), and *A. mellifera* filamentous virus (AmFV), highlighting the importance that these mites have in the transmission of diseases and raising doubts about the propagation routes of some parasites. Furthermore, in their phoretic stage mites can also affect the flight ability and therefore affect the foraging behavior of the individuals [63].

Out of the five mite species found in this study, four (*T. putrescentiae*, *P. longanalis*, *P. longipilus*, *Kuzinia* sp.) had already been reported to be associated to *B. pauloensis* in Uruguay [42]. In this case, besides the species mentioned, the presence of *P. fucorum* was noted in one wild bumble bee worker. On the other hand, *S. acarorum* was not found, maybe due to the reduced values of prevalence and intensity previously in the country [42].

The queens showed the highest number of mite species, which is reasonable since they are the only individuals in the colony that survive and make it through the winter with mites attached to their bodies [57,64,65]. Even so, Simpson's Diversity Index showed low diversity values for the mites on the three bumble bees groups. In the case of queens, the low diversity values would respond to the high intensity of the infestation (dominance) of *T. putrescentiae*.

The laboratory workers and queens showed a high number of *T. putrescentiae* individuals, which is known for its cosmopolitan distribution and its preference for high fat and/or protein contain food [66]. The nest boxes used bumble bees breeding in captivity offer an unbeatable place for this mite proliferation since nests provide an abundant amount of pollen and wax, rich in protein and fat [67]. Additionally, the confinement increases the lack of hygiene, which makes it difficult to control the presence of this mite, situation that has been reported in the laboratory breeding of other insects [51].

The mites of the genus *Kuzinia* were associated to wild workers and queens. Those mites feed exclusively on pollen, so they can find their food both in and out of the bumble bee nest, which would explain their abundance in the individuals that were foraging in the fields and their scarcity in those bumble bees that were confined to a nest [68,69]. Their presence in queens is expected since these were collected after their hibernation, when the bumble bee cycle begins promoting dispersal of the mites. *Kuzinia* mites can also be found on other bee species, wasps, beetles and other groups of insects [70].

Three different species of *Kuzinia* sp. have been described in bumble bees based on morphology (body size, shape, and number of setae in the tarsiI-IV): *K. affinis*, *K. laevis and K. Americana* [52,70]. Despite this, it is difficult to identify these mites at the species level and their taxonomy is in revision.

The two species of *Pneumolaelaps* feed directly on pollen and wax from the nests, gathering near the larvae to receive the food. Even when feeding this way, mites are heavily associated to queens [67], which matches with the results showed in this study.

Meanwhile, *P. fucorum*, is a mite of great size that feeds on pollen and small arthropods present on the bumble bee nest [68]. In this study a single specimen was found, in accordance with recent studies in where a low prevalence or even absence of this mite was noted [42,59,61].

The microsporidia *N. ceranae* and *T. pampeana* were found in the three groups of bumble bees. In Uruguay both species had already been associated to *B. pauloensis* [16,41]. The natural host of *N. ceranae* is the Asian bee *Apis cerana* Fabricius [71]. However, it was found infecting

many species of bumble bees around the world, which could impact negatively on their populations [14,30].

*Nosema ceranae* showed higher prevalence in wild workers (45.9%) than in lab workers or queens. This prevalence was different than previous studies in which a prevalence of 72% was reported in workers collected in 2010 [16] and 28.6% in 2012 [41].

No significant differences were observed in the spore counts between groups. These results do not match with those found by Plischuk *et al.* [41] in *B. pauloensis* from Uruguay, where workers were more infected than queens.

The differences in the prevalence and infection level found between different studies could be due regional differences and time of the year in which bumble bees were collected, as well as in the sampling effort. In honey bees, the prevalence of *N. ceranae* varies within the region and time of the year [72,73]. Even more, the pollen diversity available for honey bees also influence the infection level [74,75]. This issue has been barely studied in bumble bees. Rotheray *et al.* [76] found a negative relationship between *N. ceranae* infection level and the amount of food (pollen and sugar syrup) that was given to colonies of *B. terrestris*.

*Tubulinosema pampeana* was described associated to *B. pauloensis* in Argentina [54]. The prevalence of this parasite in queens, wild workers and laboratory workers was low, coinciding with previous results obtained in Argentina [54]. However, in previous study in Uruguay, Plischuk *et al.* [41] found *T. pampeana* in 36.2% of the sampled *B. pauloensis* queens and only in 1.8% of the workers, suggesting that time of the year may also influence the prevalence. Strikingly, both in Argentina and in Uruguay *T. pampeana* was only spotted in a few zones [41,54]. The impact that this new microsporidium can have at an individual or colony level is unknown. Plischuk *et al.* [54] found it infecting fat, neural and connective tissues, Malpighian tubules, muscle cells and digestive tract, so relevant effects are expected at individual level.

The nematode *S. bombi* is a parasite widely distributed throughout the world, that has been found in approximately 30 species of bumble bees [77]. In this study it was found at lower prevalence than in a previous study in the same Country [41]. This nematode has also been reported in the neighbouring Argentina [78]. Just like with microsporidia, the variations in the proportion of affected queens could be due to the site and time of the collection, and especially due to the conditions of hibernation. It can cause queen infertility and make them fly over the ground and for less time [58,77,79]. This nematode has a great incidence in the success of the laboratory breeding, since when present in a queen, it will not allow her to start a colony [78].

Diptera from Conopidae family are parasitoid with a wide distribution, which have been heavily associated to bumble bees. Its presence can trigger abnormal responses in bumble bees: they change their eating pattern, spend more time outside of the nest and exhibit a burial behavior during the last stages of parasitoidism [80,81]. In this study, larvae were found in wild workers (16.2%) and queens (10.5%). These prevalence values are superior to those found by Plischuk *et al.* (28%) [41], even though it has to be considered that in this study a lower sample size was used.

Different RNA viruses (BQCV, SBV, DWV y ABPV) were detected in Uruguayan bumble bees; over 80% of the specimens exhibited at least one of them. Those viruses are frequently found in honey bees around the world [82], including Uruguay [72,74,83]. Since they have been reported to be associated to other insects, they should be considered as multi-hosts pathogens [17,23,29].

Confinement conditions of the bumble bees during the artificial breeding did not influence the increase of the virosis, since wild workers showed higher prevalence of BQCV compared to laboratory reared workers. Wild workers may be more exposed to viral infections than laboratory workers, since in the field, bumble bees could exchange viruses with honey bees, for instance, through the flowers that both species visit. In this sense, recently Alger *et al.* [24]

found a higher prevalence of DWV and BQCV in bumble bees compared to neighbour honey bees. Even more they detected a bee virus in 19% of the flowers. These result shows how virus spillover can occur between two species that share food sources. DWV is well known in honey bees, and its association with the ectoparasitic mite *Varroa destructor* Anderson and Trueman could cause important colony losses [84,85]. Different DWV variants have been reported in honey bees, but their presence in bumble bee species and their role in the populations needs to be addressed [85–87].

## Final considerations

*A priori* it could be considered that the conditions of confinement of the bumble bees in artificial breeding colonies, together with the abundant food and the impossibility to fly would favor the proliferation of parasites and viruses. In this study this was observed in particular for mite species associated to stored foods (*T. putrescentiae*). However, wild workers showed a higher prevalence of *N. ceranae*, mites of the genus *Kuzinia*, BQCV and SBV, and higher diversity of mites, than laboratory workers. An explanation to this difference is that in the field bumble bees are in contact with parasites and viruses from honey bees or other pollinators, with the flowers acting as viral and pathogens hot spots [24]. Another factor that could explain the higher presence of parasites and viruses in the wild workers is that we collected forager bees, which can be of an older age than those bumble bees extracted from laboratory colonies. The bumble bee's age was not contemplated in this study and could be relevant. For instance, in the case of honey bees the *N. ceranae* spore count is higher in foragers than in nurses [88].

Parasites and viruses found in laboratory workers can come from two sources: the queen or the pollen the larvae were fed with (corbicular pollen from honey bees). Bumble bee queens exhibited every parasite and virus searched in this study, with a high level of infection by *N. ceranae* (although of low prevalence) and a high diversity of mites. This is expected if we consider that queens are the only individuals that survive the decay of the colony and the parasites depend in good measure of them to last until the start of a new colony [57].

The results of this study complement those carried out by Arbulo *et al.* [16], Plischuk *et al.* [41] and Revainera *et al.* [42], improving the sanitary map of the native bumble bees of Uruguay. Besides that, this study evidence that native bumble bees share several pathogens and viruses with honey bees highlighting the role of domesticated animals, which may act as reservoirs favoring the spillover to other host [9,24,30].

## Supporting information

**S1 Table. Data of parasites and virus presence.**
(PDF)

**S2 Table. Data of mites' diversity.**
(PDF)

## Acknowledgments

Authors thank the degree students Adrián Ortíz and Juan Vázquez for their collaboration in the maintenance tasks of the bumble bees in the laboratory and sample preparation stage. Authors also thank Natalia Arbulo and to anonymous reviewers for their constructing comments of the manuscript.

## Author Contributions

**Conceptualization:** Sheena Salvarrey, Matías Maggi, Ciro Invernizzi.

**Data curation:** Sheena Salvarrey.

**Formal analysis:** Sheena Salvarrey, Karina Antúnez, Santiago Plischuk, Pablo Revainera, Ciro Invernizzi.

**Funding acquisition:** Sheena Salvarrey.

**Investigation:** Sheena Salvarrey, Karina Antúnez, Santiago Plischuk, Pablo Revainera, Matías Maggi, Ciro Invernizzi.

**Methodology:** Sheena Salvarrey, Karina Antúnez, Daniela Arredondo, Santiago Plischuk, Pablo Revainera, Matías Maggi, Ciro Invernizzi.

**Project administration:** Sheena Salvarrey.

**Resources:** Sheena Salvarrey.

**Software:** Sheena Salvarrey, Karina Antúnez, Santiago Plischuk, Pablo Revainera, Ciro Invernizzi.

**Supervision:** Sheena Salvarrey, Karina Antúnez, Santiago Plischuk, Pablo Revainera, Matías Maggi, Ciro Invernizzi.

**Validation:** Sheena Salvarrey, Karina Antúnez, Daniela Arredondo, Ciro Invernizzi.

**Visualization:** Sheena Salvarrey.

**Writing – original draft:** Sheena Salvarrey, Ciro Invernizzi.

**Writing – review & editing:** Sheena Salvarrey, Karina Antúnez, Daniela Arredondo, Santiago Plischuk, Pablo Revainera, Matías Maggi, Ciro Invernizzi.

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
