## [Decision Letter · Decision Letter 0]

15 Jan 2021

PONE-D-20-36603

Parasites and RNA viruses in wild and laboratory reared bumblebees Bombus pauloensis (Hymenoptera: Apidae) from Uruguay

PLOS ONE

Dear Dr. Salvarrey,

Thank you for submitting your manuscript to PLOS ONE. After careful consideration, we feel that it has merit but does not fully meet PLOS ONE’s publication criteria as it currently stands. Therefore, we invite you to submit a revised version of the manuscript that addresses the points raised during the review process.

We look forward to receiving your revised manuscript.

Kind regards,

Guy Smagghe, PhD

Academic Editor

PLOS ONE

Journal Requirements:

Reviewers' comments:

Reviewer's Responses to Questions

**Comments to the Author**

1. Is the manuscript technically sound, and do the data support the conclusions?

Reviewer #1: Partly

Reviewer #2: Partly

2. Has the statistical analysis been performed appropriately and rigorously? 

Reviewer #1: Yes

Reviewer #2: Yes

3. Have the authors made all data underlying the findings in their manuscript fully available?

Reviewer #1: Yes

Reviewer #2: No

4. Is the manuscript presented in an intelligible fashion and written in standard English?

Reviewer #1: No

Reviewer #2: No

5. Review Comments to the Author

Reviewer #1: Given the topic and type of this paper, the data and analyses done are sufficient, but overall I miss objectives which make this paper relevant and worthwhile.

I decided to put 'minor revisions' because the objectives, although they are very minimal, are achieved. But i do strongly suggest to read my comments and answer some questions, and I hope that they will be helpful.

The heterogeneity of the types of pathogens screened for is good, but I do miss some additional analysis.

For example, the choice of only microscopic screening of internal pathogens does not make the methods very sensitive. Only two microsporidian species were chosen to screen for. Why the choice of these parasites? Why not other internal parasites like Crithidia sp. or Apicystis sp.? This might give a lot more interpretable data, and might give enough parasite diversity data to actually state some objectives and get some more information regarding parasites in bumblebees, rather than only report the findings. I also wonder how did you distinguish between Nosema species? How did you see a difference between N. ceranae and N. bombi without molecular confirmation?

For me the sampling set up is not very clear: why take queens from wild and rear them, and then compare them with workers from the wild but from another year? What was the original motive to do this,

[199] having a similar prevalence between queens and laboratory workers is not so strange, since they don’t come into contact anymore with the environment, the lab workers can only get parasites from the queens; the queens already infested will transmit their parasites, while the queens not infested cannot transmit parasites, so the prevalence will stay similar.

What would be a relevant analysis, is to compare the parasite diversity between the queens and the lab workers; are there any parasites that are not present (or way less) in the lab workers but were omnipresent in the queens? For example from your data, there doesn’t seem to be a bottle neck for Nosema ceranae, but a small bottleneck for T. pampeana. Even better would be to also analyse the parasites of the queens used for rearing, if they were kept frozen after the experiment.

[255] You mention mites being a possible vector, but you don’t analyse this in your data; for this you don’t necessarily have to screen the mites themselves, but you could model a relation between presence of certain mite species and prevalence/abundance/load of viruses and pathogens.

Some smaller comments:

The general structure is not good yet. Some examples:

55 strange sentence ‘another example of pathogen spillover are RNA viruses’. Then you explain how viruses found in honeybees are also found in other bee species, but you don’t provide any context about why these viruses are considered to be a spillover risk.

45 ‘among others’ fits better than ‘between others’. Also, maybe it is good to mention multiplicative effects, like pesticide intoxication that increases pathogen susceptibility.

85-87 I do not understand this sentence. This sentence seems to state that B. terrestris and .B ruderatus is present in Uruguay, while on line 72 you say there are only B. pauloensis and B. bellicosus. Also the second part of the sentence on 86 does not make sense.

94 you mention 33 queens, and then explain how many are used for different analysis; then you mention 40 (of these 33??) were used to start laboratory rearing. I don’t think that is physically possible… Also were the queens collected from a lab experiment or were they actually caught during foraging from the wild? This is not explained clearly.

There are still a lot of grammar and spelling mistakes in the text. Some examples:

84 ‘significant’ damage

107 grammatical mistake

287-289 strange sentence

327 spotted

392 strange sentence

397 strange sentence and weird conclusion?

Reviewer #2: The authors did not make all data available. There were not Supporting Information or Supplementary Material.

The English language is ok. However, there were shown typographical/grammatical errors (minimum). I am not English language specialist; therefore, I can not question the writing of the manuscript.

6. PLOS authors have the option to publish the peer review history of their article (what does this mean?). If published, this will include your full peer review and any attached files.

Reviewer #1: **Yes: **Tina Tuerlings

Reviewer #2: No

---

## [Author Response · Author response to Decision Letter 0]

1 Mar 2021

Response to reviewers

Reviewer #1:

The heterogeneity of the types of pathogens screened for is good, but I do miss some additional analysis. For example, the choice of only microscopic screening of internal pathogens does not make the methods very sensitive. Only two microsporidian species were chosen to screen for. Why the choice of these parasites? Why not other internal parasites like Crithidia sp. or Apicystis sp.? This might give a lot more interpretable data, and might give enough parasite diversity data to actually state some objectives and get some more information regarding parasites in bumblebees, rather than only report the findings.

R: We used the microscopical screening to search for the presence of several parasites that can be identified with this method (e.g. Kinetoplastidea, Neogregarinorida and Microsporidia). The presence of Crithidia sp. and Apycistis sp. was also evaluated but those parasites were not found. This information was included in the new version of the manuscript. 

… I also wonder how did you distinguish between Nosema species? How did you see a difference between N. ceranae and N. bombi without molecular confirmation? 

R: We are convinced that the spores were N. ceranae since I) previous studies carried out by molecular methods have never found N. bombi in Bombus spp. in Uruguay. Closest reports were 1,400 km far from the country (e.g.: Plischuk, 2013; Schmid-Hempel et al., 2014; Arbulo et al. 2015; Plischuk & Lange, 2016; Plischuk et al., 2017a, b), and II) careful dissections of all bees showed that spores were always infecting the gut (target tissue of N. ceranae) and they never were found into Malpighian tubules (target tissue of N. bombi).

For me the sampling set up is not very clear: why take queens from wild and rear them, and then compare them with workers from the wild but from another year? What was the original motive to do this?

R: We collected queens from the wild in spring 2014 (September in south hemisphere) and some of them were used for the parasites and viral analysis and others for the rearing in laboratory. Wild workers were collected in autumn (March 2015) since it is the only time of the year when they are found in great numbers in the wild, in accordance with their biological cycle. It is not possible to find many wild workers in spring. Spring 2014 and autumn 2015 belong to the same “biological year” for this species. We compared the workers reared in the lab with the workers and queens collected in the wild.

having a similar prevalence between queens and laboratory workers is not so strange, since they don’t come into contact anymore with the environment, the lab workers can only get parasites from the queens; the queens already infested will transmit their parasites, while the queens not infested cannot transmit parasites, so the prevalence will stay similar. What would be a relevant analysis, is to compare the parasite diversity between the queens and the lab workers; are there any parasites that are not present (or way less) in the lab workers but were omnipresent in the queens? For example, from your data, there doesn’t seem to be a bottle neck for Nosema ceranae, but a small bottleneck for T. pampeana. Even better would be to also analyse the parasites of the queens used for rearing, if they were kept frozen after the experiment.

R: We agree with your comments and we considered that some of those ideas are already included in the manuscript. As the example that you mentioned about the two microsporidia in the discussion section we discuss about other examples such as a P. longanalis, and SBV (Sacbrood virus). On the other hand, unfortunately the queens used for rearing were not kept frozen since laboratory colonies were used for pollination experiments; we are aware that including them in the analysis would have enrich this study.

[255] You mention mites being a possible vector, but you don’t analyses this in your data; for this you don’t necessarily have to screen the mites themselves, but you could model a relation between presence of certain mite species and prevalence/abundance/load of viruses and pathogens.

R: Its very interesting what you mention about modeling but unfortunately, we don’t have the necessary data (e.g. load) for all groups of pathogens and viruses to do that. 

Some smaller comments: The general structure is not good yet. Some examples:

55 strange sentence ‘another example of pathogen spillover are RNA viruses’. Then you explain how viruses found in honeybees are also found in other bee species, but you don’t provide any context about why these viruses are considered to be a spillover risk.

R: We modified the text and your suggestions were included. 

45 ‘among others’ fits better than ‘between others’. 

R: The text was modified taking into account this observation.

Also, maybe it is good to mention multiplicative effects, like pesticide intoxication that increases pathogen susceptibility.

R: It was mentioned in the revised version. 

85-87 I do not understand this sentence. This sentence seems to state that B. terrestris and .B ruderatus is present in Uruguay, while on line 72 you say there are only B. pauloensis and B. bellicosus. Also the second part of the sentence on 86 does not make sense. 

R: The text was modified taking into account these observations.

94 you mention 33 queens, and then explain how many are used for different analysis; then you mention 40 (of these 33??) were used to start laboratory rearing. I don’t think that is physically possible… Also were the queens collected from a lab experiment or were they actually caught during foraging from the wild? This is not explained clearly.

R: We corrected the numbers of the total collected queens and added details to the material and methods section to improve the clarity.

There are still a lot of grammar and spelling mistakes in the text. Some examples:

84 ‘significant’ damage

107 grammatical mistake

287-289 strange sentence 

327-spotted

392-strange sentence

397 strange sentence and weird conclusion?

R: The manuscript was revised and improved, taking into account all the suggestions.

Reviewer 2

Line 2- Bumble bees

R: It was corrected. 

Line 20- Please change bumblebees by bumble bee. I recommend making this change in the full text. Also in the case of honeybees by honey bees.

R: It was corrected.

Line 35- bumble bees 

R: It was corrected.

Line 45- Please, to continue in the Line 45. ...between others (2). In particular, ....

 R: It was corrected.

Line 49- Sentence in Lines 49-51, tend to ensure that Apis mellifera is main agent of dispersion of pathogens to other insect species, which is not clear at all. Please reconsider the setence. I recomend to consider the honey bees as one of posible agent of pathogen dispersion. ....while others have a broad host spectrum [6,7]. There is e.g. that the extended commerce and movement of managment bees such as honey bees, has led .... 

R: We agree that A. mellifera is not the only agent of pathogens dispersion but that is not the intention of the reference sentence. We just wanted to highlight the role of commercial colonies as pathogens and parasites reservoirs basing this idea on several references (6,8,9,10,11). We made little modification in the text to clarify the idea, including bumble bees among the commercial colonies. 

Line 54- Bumble bees 

R: It was corrected.

Line 50- Apis mellifera add “L.” 

R: It was corrected.

Line 52- Nosema ceranae add “Fries” 

R: It was corrected.

Line 62- Friese 

R: It was corrected.

Line 67- Move to Line 88. 

From a sanitary point of view, the artificial breeding conditions (high density of individuals, impossibility of going out to defecate and forage, and limited food availability), can increase the survival and multiplication of different pathogens, facilitating the proliferation and transmission of diseases [39,40]. Thus, the aim the this study was perform an exhaustive survey... 

R: The text was modified as suggested.

Line 72- Smith 

R: It was corrected.

Line 75- To include species descriptor 

R: It was corrected.

Line 76- Dufour 

R: It was corrected.

Line 77- To include Descriptor of these species

R: It was corrected.

Line 81- L. and F. 

R: It was corrected.

Line 82- Guérin-Méneville 

R: It was corrected.

Line 85- I recommend to read and to include Arismendi et al 2021. Occurrence of bee viruses and pathogens associated with emerging infectious diseases in native and non-native bumble bees in southern Chile. Biological Invasion https://doi.org/10.1007/s10530-020-02428-w

R: The paper was cited and included in the references (47).

Line 96- Why did not you use all the samples for the viral, ectoparasites and other pathogens analysis? That is, the 33 samples. I believe that these pathogens/parasites can be detected in one sample. 

How did you maintain the samples? in ethanol? under cold condition (-20°C/-80°C). Please explain/answer in details these questions

R: It was not possible to perform all analysis in the same individuals since every parasite/ pathogen requires different conditions of sample storage (-20 or -80°C) and different methods for sample processing. For example, samples for viral analysis requires storage at -80°C and the complete specimen is homogenized for RNA extraction. On the other hand, samples for internal parasites analysis require storage at -20°C, specimens must be dissected, the midgut extracted and observed. Procedures are not compatible to be performed in the same individual.

Line 97- There is not clear!

How many queens were collected after hibernation period? 33 or 73?. 

I feel that, there were collected 73 queens (see Lines 94-97). In which case, 33 were used for pathogen/parasites analisys and 40 queens were used to start rearing under laboratory condiction.

Please clarify!

R: In total 73 queens were collected, 33 for analysis and 40 for rearing. It was clarified in the text.

Line 99- Similar to above comments, why not use the 92 worker bumble bees for internal and external pathogen/parasites? 

R: As explained before, it was not possible to perform all analysis in the same individuals since every parasite/ pathogen requires different conditions of sample storage (-20 or -80°C) and different methods for sample processing. 

Line 113- to continue in the Line 112. 

R: It was corrected.

Line 138- No italic 

R: It was corrected.

Line 139- No italic 

R: It was corrected.

Line 141-add dut 

R: It was corrected.

Line 150- No italic

R: It was corrected.

Line 151- ). 

R: It was corrected.

Line 152- In the Table 1, only β-actin was listed. Did you also used RPS5? 

R: Only β-actin was used. It was corrected in the text.

Line 160- Please indicate the statistical software. 

R: InfoStat software was used. It was included in the text.

Line 172- Please add real p value. In this case must be report as: Chi-square: = 7.52; p =0.006; df = 1). 

R: It was corrected

Line 174- Is it the real p-value? I believe that the p-value is p <001. Please clarify! 

R: It was corrected

Table 2- Please indicate N° samples

Indicate N° samples

Where is the wild queens values?

You stated in the 

Lines 96-97 that 33 queens were collected for pathogens/parasites analysis

R: The text was modified as suggested. 

Line 180- p = 0003 

R: The real p value was added

Line 182- co-infestation 

R: It was corrected.

Line 187- Please to include title in the Y-axis in the Figure 1, anexed in this manuscript. 

R: It was corrected.

Line 191- Delete Lines 191-192. To include this information at end of Figure 1. 

R: We think it should be explained in the figure legend to understand the figure. 

Line 198-real p-value 

R: It was included. 

Line 199-real p-value 

R: It was included. 

Line 202, 203, 207, 208- change “individual” for “bee”

R: It was corrected.

Line 220- Real p-value 

R: It was included. 

Line 223- I recommend to include ct values in supplementary material 

R: If the reviewer think it is essential, we can include it. However, we think it does not provide substantial information to the manuscript.

Line 227- Real p-value 

R: It was included. 

Line 228- Real p-value 

R: It was included. 

Line 229- Please to include title in Y-axis 

R: It was included. 

Please clarify the asteristic in the bars. Who or what is it being compared to? You must to be illustrative and clear to the reader.

R: It was corrected

Line 232- Move this setence at end of Figure 2. 

R: We think this information should be in the figure legend. 

spell asterisk. No symbol. 

The asterisk indicates ...

R: It was corrected.

 Line 235-238- There were not shown the viral load. 

I am confused. If you used Real Time PCR, there exist the posibility to make a viral quantification. Futhermore, you cited Pfaffl 2001.

Pfaffl W. A new mathematical model for relative quantification in real-time 590 RT–PCR. Nucleic Acids Res. 2001;425(3):2002–7.

If you did not quantified the viral load in the infected bumble bees, I recommend to do it. I believe this information could be usefull to related the viral prevalence with viral load. One thing is prevalence (positive cases) and the other, is the infection level (high, medium or low load of viral infection). There exist the posibility that hight prevalence could be related to high viral load. On the hand, low prevalence, could be associated to low viral titer. The viral load could have implicance in viral dispersion between bee species. 

Of course that the viral prevalence is importante index, however, without viral load, it is difficult to conceptualize the intensity of infection or exposure of infected bees. This provides important information on if bee viruses are present in bumble bees at levels likely to have consequences for their health. 

For more details, please see:

Dolezal AG, Hendrix SD, Scavo NA, Carrillo-Tripp J, Harris MA, Wheelock MJ, et al. (2016) Honey Bee Viruses in Wild Bees: Viral Prevalence, Loads, and Experimental Inoculation. PLoS ONE 11(11): e0166190. doi:10.1371/journal. pone.0166190

Alger SA, Burnham PA, Boncristiani HF, Brody AK (2019) RNA virus spillover from managed honeybees (Apis mellifera) to wild bumblebees (Bombus spp.). PLoS ONE 14(6): e0217822. https://doi.org/10.1371/journal.

pone.0217822

Alger SA, Burnham PA, Brody AK (2019) Flowers as viral hot spots: Honey bees (Apis mellifera) unevenly deposit viruses across plant species. PLoS ONE 14(9): e0221800.

 https://doi. org/10.1371/journal.pone.0221800

Arismendi et al 2021. Occurrence of bee viruses and pathogens associated with emerging infectious diseases in native and non-native bumble bees in southern Chile. Biological Invasion https://doi.org/10.1007/s10530-020-02428-w

R: β-actin mRNA was amplified in each sample as a control of correct RNA manipulation and extraction. The text was modified accordingly. The reference Pfaffl 2001 was not adequate, since viral quantification was not carried out.

We agree that absolute quantification would provide interesting information and will allow the comparison between viral levels in honey bees and bumble bees. However, at this point it is not possible to perform since we did not include adequate standard curves (dilution series of cloned PCR products or commercial gene fragments, for each virus). 

In spite of that, we think that information regarding RNA viruses prevalence, together with the information of other parasites and pathogens provides useful information.

Line 244- This is the objective. It should be in Lines 88-90. 

R: It was corrected.

Line 246- Delete. It does not provide relevant information. 

R: It was deleted.

Line 249- I recommend to delete the subtitles in Discussion section 

R: Subtitles were deleted.

Line 250- ... phoretic mites in bumble bee... There are tracheal mites which to be harmful for bumblebees. 

R: It was corrected. 

Line 254- and 

R: It was added

Line 273- Please to include reference! 

R: It was included. 

Line 275- To include reference 

R: It was included. 

66 - Koulianos, S., Schwarz, H.H. (1999) Reproduction, development and diet of Parasitellus fucorum (Mesostigmata: Parasitidae), a mite associated with bumblebees (Hymenoptera: Apidae). J. Zool., 248, 267–269.

67- Royce, L.A., Krantz, G.W. (1989) Observations on pollen processing by Pneumolaelaps longanalis (Acari, Laelapidae), a mite associate of bumblebees. Exp. App. Acarol., 7, 161–165.

Line 287- To include comma (,) 

R: It was corrected.

Line 298- Delete

 R: It was deleted.

Line 301- ...Apis cerana (70). However, 

R: It was corrected

Line 310- may or could?

 R: It was modified as suggested.

Line 316- Even more,

 R: It was modified as suggested.

Line 317- honey bee 

R: It was modified as suggested.

Line 324- Uruguay 

R: It was modified as suggested.

Line 332- Delete 

R: It was modified as suggested.

Line 344- Delete 

R: It was modified as suggested.

Line 353- Delete 

R: It was modified as suggested.

Line 358- These data are results. 

R: It was modified as suggested.

Line 361- Viral load data were not shown in the manuscript. 

R: Viral load was not estimated. The text was modified to viral prevalence.

Line 378- no italic 

R: It was corrected.

Line 400- Add Acknowledgments and Author Contributions

R: It was included according to journal guidelines

Line 401- References

R: It was corrected.

---

## [Decision Letter · Decision Letter 1]

26 Mar 2021

Parasites and RNA viruses in wild and laboratory reared bumblebees <bombus pauloensis=""> (Hymenoptera: Apidae) from Uruguay

PONE-D-20-36603R1</bombus>

Dear Dr. Salvarrey,

We’re pleased to inform you that your manuscript has been judged scientifically suitable for publication and will be formally accepted for publication once it meets all outstanding technical requirements.

Kind regards,

Guy Smagghe, PhD

Academic Editor

PLOS ONE

Additional Editor Comments (optional):

Reviewers' comments:

Reviewer's Responses to Questions

**Comments to the Author**

1. If the authors have adequately addressed your comments raised in a previous round of review and you feel that this manuscript is now acceptable for publication, you may indicate that here to bypass the “Comments to the Author” section, enter your conflict of interest statement in the “Confidential to Editor” section, and submit your "Accept" recommendation.

Reviewer #1: All comments have been addressed

2. Is the manuscript technically sound, and do the data support the conclusions?

Reviewer #1: Yes

3. Has the statistical analysis been performed appropriately and rigorously? 

Reviewer #1: Yes

4. Have the authors made all data underlying the findings in their manuscript fully available?

Reviewer #1: Yes

5. Is the manuscript presented in an intelligible fashion and written in standard English?

Reviewer #1: Yes

6. Review Comments to the Author

Reviewer #1: (No Response)

7. PLOS authors have the option to publish the peer review history of their article (what does this mean?). If published, this will include your full peer review and any attached files.

Reviewer #1: **Yes: **Tina Tuerlings

---

## [Editor Report · Acceptance letter]

14 Apr 2021

PONE-D-20-36603R1 

Parasites and RNA viruses in wild and laboratory reared bumble bees *Bombus pauloensis* (Hymenoptera: Apidae) from Uruguay 

Dear Dr. Salvarrey:

I'm pleased to inform you that your manuscript has been deemed suitable for publication in PLOS ONE. Congratulations! Your manuscript is now with our production department. 

Kind regards, 

on behalf of

Prof. Guy Smagghe 

Academic Editor

PLOS ONE